# Assessing the Nutrient Status of Low Carbohydrate, High-Fat (LCHF) Meal Plans in Children: A Hypothetical Case Study Design

**DOI:** 10.3390/nu14081598

**Published:** 2022-04-12

**Authors:** Caryn Zinn, Kayla-Anne Lenferna De La Motte, Amy Rush, Rebecca Johnson

**Affiliations:** 1Human Potential Centre, School of Sport & Recreation, Faculty of Health & Environmental Sciences, Private Bag 92006, Auckland 1142, New Zealand; kaylahaycock22@gmail.com; 2Type 1 Diabetes Family Centre, 11 Limosa Close, Stirling 6021, Australia; amy@type1familycentre.org.au (A.R.); bec@type1familycentre.org.au (R.J.)

**Keywords:** low-carbohydrate, high-fat, LCHF, children, adolescents, nutrient reference values, NRVs

## Abstract

There is well-established evidence for low-carbohydrate, high-fat (LCHF) diets in the management of chronic health conditions in adults. The natural next step is to understand the potential risks and benefits of LCHF diets for children, where they may have useful applications for general health and a variety of chronic health conditions. It is vital that any diet delivers sufficient micronutrients and energy to ensure health, wellbeing, and growth. This descriptive study assesses the nutrient and energy status of LCHF sample meal plans for children. We designed four meal plans for hypothetical weight-stable male and female children (11 years) and adolescents (16 years). Carbohydrates were limited to ≤80 g, protein was set at 15–25% of the total energy, and fat supplied the remaining calories. Using FoodWorks dietary analysis software, data were assessed against the national Australian/New Zealand nutrient reference value (NRV) thresholds for children and adolescents. All meal plans exceeded the minimum NRV thresholds for all micronutrients; protein slightly exceeded the AMDR recommendations by up to three percentage points. This study demonstrates that LCHF meal plans can be energy-, protein-, and micronutrient-replete for children and adolescents. As with any dietary approach, well-formulated meals and careful planning are key to achieving the optimal nutrient status.

## 1. Introduction

Carbohydrate restriction is increasingly being used in adult populations to manage health conditions, including overweight and obesity [1] and both type 1 and type 2 diabetes [2,3]. The evidence indicates that low-carbohydrate, high-fat (LCHF) diets may also be effective for the management of metabolic conditions in children and adolescents [4,5,6,7]. With any dietary intervention that involves restricting certain foods, it is important to be confident the approach can deliver adequate nutrients and energy. This is particularly important to assure growth and development in paediatric patients.

To date, the application and efficacy of certain LCHF diets in children and adolescents have largely been explored in very specific contexts, for example, versions of a ketogenic diet as a treatment method for children with intractable epilepsy [8,9,10,11,12]. The ketogenic diets used in the treatment of epilepsy have a very high proportion of fat compared to protein and carbohydrates (for example, 90% of calories derived from fat, 8% from protein, and 2% from carbohydrates) [13]. It is highly unlikely that such a macronutrient distribution would allow for an adequate nutrient supply, and as such, extreme versions of LCHF diets such as this are warranted only as therapeutic interventions for specific medical conditions, and micronutrient supplementation will be necessary in these circumstances.

However, the nutrient density of less extreme versions of an LCHF diet for children is worth exploring, because lower-carbohydrate approaches are increasingly preferred by the general population. Currently, in accordance with messages from dietitians, parenting experts, and the media, many families are shifting away from processed foods for general health reasons, which naturally reduces their carbohydrate intake. It is therefore vital to understand the nutrient status of diets that contain fewer carbohydrates and whether they can be appropriate for younger people. While LCHF approaches have been explored in children and adolescents to manage obesity and diabetes, one recent study considered low-carbohydrate diets for obese adolescents [14], and another considered LCHF diets in the context of glycaemic control in children with type 1 diabetes, for example [15]—an assessment of the nutritional adequacy of LCHF diets intended for healthy children of normal weight has not yet been done.

Our previous work demonstrated that, for adults, an LCHF diet can be micronutrient- and energy-replete if it is well-planned [16]. The purpose of this work is to assesses the macro- and micronutrient analyses of moderately low-carbohydrate meal plans for children and adolescents of healthy weights, planned with children’s taste preferences and palates in mind. For comparison, we also analysed the nutrient status of a sample meal plan designed for a 9–11-year-old child and endorsed by the Australian Dietary Guidelines government group [17].

## 2. Materials and Methods

In this descriptive study, we formulated and analysed four one-day LCHF meal plans for male and female children and adolescents. Our approach focused on reducing the intakes of carbohydrate-rich foods such as confectionary, sugary drinks, bread and other baked goods, breakfast cereals, pasta, rice, and some fruits, replacing them with lower-carbohydrate alternatives. Our meal plans generally feature vegetables; nuts and seeds; dairy products; and proteins such as meat, poultry, and eggs. We chose to create plans for children aged 11 and 16, as these ages are the midpoints for the Nutritional Reference Value (NRV) age ranges of 9–13 years and 14–18 years [18]. We used weight, height, and BMI averages from WHO BMI and height growth charts to ascertain the reference weights, heights, and BMI of each of the hypothetical subjects [19]. To estimate the total energy expenditure, we used the Schofield equation, inputting the weight and height variables we had established earlier and an activity factor of 1.7 (light–moderate level) [20]. We set the total energy intake required in our meal plans in accordance with the energy needs of each of the four hypothetical subjects (±5%). We refined the 9–11-year-old children demographic data presented with the sample ADG diet of a ten-year-old (midpoint) female. We obtained weight, height, BMI, and energy intake using the same methods described above to undertake our dietary analysis [17]. Table 1 presents the demographic data used for the LCHF case studies and the ADG sample meal plan.

There are no universally agreed upon macronutrient ratios for a LCHF diet in children or adults; the most-cited definition is from Feinman et al., who defined ‘low-carbohydrates’ for adults as <130 g per day (or <26% of their total energy) [21]. The standard dietary carbohydrate guidelines for children are 45–65% of the total energy intake [18]. For this paper, we opted to select our carbohydrate content in grams rather than as a percentage of the total energy. We selected the 70–80 g range and considered it ‘low carbohydrate’, because it is placed at the midpoint between Feinman’s cut-off point (for adults) of <130 g CHO/day and that of the upper threshold of very low-carbohydrate or ketogenic diets, such as those used therapeutically in the treatment of epilepsy (i.e., 50 g/day). We also considered convenience, palatability, and acceptability in setting our carbohydrate target. The protein threshold was calculated based on the acceptable macronutrient distribution range (AMDR) as per the Australian NHMRC and New Zealand MOH guidelines: 15–25% of total energy intake. The remaining energy required by each subject came from fat sources.

We created each meal plan with the aim of achieving at least 100% of the NRV for each micronutrient according to the Recommended Dietary Intakes (RDIs) or Adequate Intakes (AIs) if the RDI was not available [18]. Each of the four meal plans were developed using a predominantly whole-food principle (i.e., using minimally processed foods) as a foundation. Foods that were generally considered popular and acceptable options for children (according to the three nutrition professionals on our team who regularly work with children) were included, and those that were not (i.e., fish or boiled eggs packed into school lunch boxes) were intentionally excluded. We used computer analysis software FoodWorks 10, Professional edition. Version 10.0.4262 (Xyris software, Australia Pty Ltd.), which uses an Australian and New Zealand food database, to analyse our meal plans and that of the sample meal plan provided by the governmental group that issue the ADG for children. No patients were involved in this work.

## 3. Results

The LCHF sample meal plans and their corresponding nutrient analyses for the 11-year-old and 16-year-old males and females are presented in Table 2 and Table 3, respectively. All four meal plans met the energy requirements and exceeded the NRV thresholds for all micronutrients. The protein amounts were adequate; each meal plan slightly exceeded the AMDR recommendations (by up to three percentage points).

**Table 2 nutrients-14-01598-t002:** LCHF sample meal plans.

LCHF Meal Plan 1 (Female; 11 Years)	LCHF Meal Plan 2 (Male; 11 Years)	LCHF Meal Plan 3 (Female; 16 Years)	LCHF Meal Plan 3 (Male; 16 Years)
Breakfast *Low-carb granola* 5 raspberries, 50 g Purely Elizabeth granola, 200 g Greek yoghurt	Breakfast *Eggs on toast* 1 slice Helga’s lower carb five seed bread, 2 tsp salted butter, 2 regular boiled eggs, and ½ tsp iodised table salt	Breakfast *Smoothie* 80 g strawberries, 1 tb peanut butter, 5 brazil nuts, ½ cup Greek yoghurt, and ½ cup tap water	Breakfast *Eggs on toast* 1 slice Helga’s lower carb five seed bread, 2 tsp salted butter, 2 regular boiled eggs, ¼ cup baked beans, and ½ cup cheddar cheese.
Morning tea *Vegetable sticks and cream cheese* 6 snow peas, 40 g red capsicum, and ¼ cup cream cheese	Morning tea *Vegetable sticks and cheese* 40 g cherry tomatoes, 40 g telegraph cucumber, 6 snow peas, 49 g cheddar cheese	Morning tea *Vegetable sticks and cream cheese* 50 g telegraph cucumber, 10 snow peas, ½ tsp iodised table salt, and ½ cup cream cheese	Morning tea *Fruit and cheese* ½ medium apple and 63 g cheese
Lunch *Frittata* 2 regular eggs, 50 g bacon, 40 g red capsicum, ½ cup baby spinach, 20 g red onion, ¼ cup cheddar cheese, 2 tsp salted butter, and ½ tsp iodised table salt	Lunch *Low-carb wrap* 1 Mountain bread natural wrap, 20 g telegraph cucumber, ½ cup spinach, 45 g avocado, 100 g chicken, ¼ cup grated cheddar cheese, and 2 tsp mayonnaise	Lunch *Frittata* 2 regular eggs, 50 g grilled chicken breast, 45 g zucchini, ½ cup baby spinach, ¼ cup red capsicum, 1 mushroom, ¼ cup brown onion, 20 g cheddar cheese, and 1 tsp olive oil	Lunch *Low-carb sandwich* 5 slices salami, 2 slices Helga’s lower carb five seed bread, 2 tsp salted butter, 1 tb mayonnaise, 40 g avocado, ½ cup baby spinach, 20 g telegraph cucumber, 21 g cheddar cheese, and 30 g tomato
Afternoon tea *Seed crackers with peanut butter* 2 tsp peanut butter and 2 Olina’s seeded crackers	Afternoon tea *Fruit, crackers, and peanut butter* ½ small apple, 2 Olina’s seeded crackers, and 1 and ¼ tb peanut butter	Afternoon tea *Mashed avocado and vegetable sticks* 80 g avocado, 1 cup carrots,	Afternoon tea *Olina’s seeded crackers with peanut butter* 2 Olina’s seeded crackers and 2 tsp peanut butter
Dinner *Chicken stir fry* 2 medium zucchinis, ½ cup broccoli, ¼ cup brown onion, 150 g chicken breast, 1 tb olive oil	Dinner *Steak, cauliflower mash, and vegetables* 150 g steak fillet, 100 g cauliflower, 2 tb cream, 100 g carrots, 50 g green beans, ½ cup peas, 1 tsp mustard, ½ tsp iodised table salt, and 2 tsp olive oil	Dinner *Beef stir fry* 175 g beef strips, 1 cup red cabbage, ½ cup carrots, ½ cup green beans, ¼ cup corn kernels, 1 tb olive oil	Dinner *Pork chops, cauliflower mash, and vegetables* 2 small pork chops, 200 g cauliflower, 2 tsp cream, ½ cup carrots, and 1 cup green beans
Supper *Yoghurt and berries* 150 g Greek yoghurt, ½ cup blueberries, 50 g strawberries	Supper *Yoghurt and berries* 150 g Greek yoghurt and 80 g strawberries	Supper *Yoghurt with nuts and seeds* 150 g Greek yoghurt, 2 tb sunflower seeds, 1.5 tb chia seeds, 1.5 tb cashews	Supper *Yoghurt with nuts* 150 g Greek yoghurt with 15 g almonds and 15 g brazil nuts

**Table 3 nutrients-14-01598-t003:** Nutrient analysis of the LCHF meal plans.

	Female Meal Plans				Male Meal Plans			
Nutrient	11 Year Old	NRV/Goal	16 Year Old	NRV/Goal	11 Year Old	NRV/Goal	16 Year Old	NRV/Goal
Energy (calories)	2077	2031.55	2351	2425.9	2129.8	2234.7	3062	2987.6
Carbohydrate (g) % TE	78.6 12	229–330 45–65	73.3 12	273–394 45–65	69.7 12	251–363 45–65	76 10	336–485 45–65
Total Sugar ^‡^ (g)	61.8	-	58.3	-	38.9	-	40.6	-
Free sugar (g) % TE	0.9 0.2	- <5%	1.2 0.2	- <5%	1.9 0.4	- <5%	2.9 0.4	- <5%
Starch (g)	16.8	-	15	-	30.8	-	35.4	-
Protein (g) % TE	139.2 27	15–25	140 24	15–25	151.7 28	15–25	175.6 23	15–25
Fat (g) % TE	128.2 55	20–35	157.1 59	20–35	142.7 57	20–35	220.7 64	20–35
Saturated fat (g) % TE	43.6 19	22.6 ≤10	57.9 22	27 ≤10	55.1 22	25 ≤10	85.5 25	33 ≤10
Trans fats (g) % TE	1.9 0.8	2.5 <1 *	2.7 1	3 <1 *	3.0 1.2	2.5 <1 *	3.2 0.9	3.3 <1 *
MUFA (g) % total fat	34.9 27	- -	64.4 42	- -	50 38	- -	78.5 36	- -
PUFA (g) % total fat	5.5 4.4	- -	22.4 15	- -	13.7 10	- -	30.0 14	- -
Linoleic acid (O6 PUFA) (g)	4.5	8 ^†^	16.97	8 ^†^	10.28	10 ^†^	23.82	12 ^†^
Alpha-linoleic acid (O3 PUFA) (g)	0.5	0.8 ^†^	4.74	0.8 ^†^	1.5	1.0 ^†^	2.74	1.2 ^†^
Omega-6: omega-3 ratio	9	10	3.6	10	6.9	10	8.7	10
Fibre (g)	28.2	20 ^†^	41.5	22 ^†^	30.5	24 ^†^	35.2	28 ^†^
Thiamine (mg)	0.99	0.9	1.87	1.1	1.02	0.9	2.66	1.2
Riboflavin (mg)	2.66	0.9	2.49	1.1	1.66	0.9	2.32	1.3
Niacin (mg)	33.4	12	24.01	14	19.47	12	30.67	16
Vitamin C (mg)	373.8	40	229.56	40	109.81	40	66.25	40
Vitamin A (µg)	1206.9	600	4490.03	700	2320.54	600	2084.79	900
Vitamin E (mg)	14.66	8 ^†^	19.54	8 ^†^	20.07	9 ^†^	24.73	10 ^†^
Vitamin B_12_ (µg)	3.16	1.8	6.72	2.4	8	1.8	7.05	2.4
Folate, total (µg)	533.67	300	694.36	400	637.23	300	684.07	400
Calcium (mg)	1198	1000	1318.83	1300	1165.10	1000	1824.58	1300
Iron (mg)	8.05	8	15.59	15	11.04	8	11.68	11
Magnesium (mg)	330.06	240	546.84	360	359.82	240	503.09	410
Zinc (mg)	9.95	6	23.62	7	20.40	6	19.45	13
Sodium (mg)	3245.94	400–800 ^†^	2524.44	460–920 ^†^	4029.34	400–800 ^†^	4242.04	460–920 ^†^
Potassium (mg)	3861.57	2500 ^†^	5158.43	2600 ^†^	3372.33	3000 ^†^	3654.74	3600 ^†^
Phosphorus (mg)	1879.67	1250	2300.29	1250	1866.57	1250	2413.19	1250
Selenium (µg)	80.76	50	222.98	60	108.25	50	320.42	70
Iodine (µg)	254.9	120	252.23	150	399.94	120	177.65	150

* WHO recommendation for trans fats. ^†^ AIs used as RDIs were unavailable. ^‡^ Total sugar is defined as ‘free sugar’ (monosaccharides and disaccharides added to foods and beverages by the manufacturer, cook, or consumer and sugars naturally present in honey, syrups, fruit juices, and fruit juice concentrate) [22] and ‘intrinsic sugar’ (natural sugar found in intact fruit, vegetables, and milk). AI, adequate intake; LCHF, low-carbohydrate, high-fat diet; MUFA, monounsaturated fat; NRV, nutrient reference value; PUFA, polyunsaturated fat; RDI, recommended daily intake; TE, total energy.

The ADG sample meal plan and the corresponding analysis is presented in Table 4. Overall, this diet fell short of the daily estimated energy requirements for a 10-year-old female by more than a third. It meets the macronutrient recommendations contained in the Australian dietary guidelines and most of the micronutrients, with Vitamin C and selenium falling marginally short at 77%, and 97%, respectively, of the RDI recommendations.

**Table 4 nutrients-14-01598-t004:** Australian Dietary Guidelines sample meal plan and nutrient analysis for a 10-year-old female.

Sample Meal Plan		
Breakfast	1 wheat biscuit, ½ cup reduced fat milk, 100 g yoghurt	
Morning snack	1 medium banana; 3 crispbreads, 1 tb of peanut butter spread	
Lunch	*Sandwich*: 2 × slices of wholemeal bread, 1 boiled egg, 1 slice reduced fat cheese (20 g), 1 cup mixed salad	
Afternoon snack	1 crumpet with 1 tsp margarine; 250 mL reduced fat milk	
Dinner	*Lamb kebab with vegetables* (65 g cooked lamb kebab, 1 small, boiled potato, ½ cup cooked carrot, ½ cup cooked beans)	
Evening snack	1 cup mixed fruit plus 100 g natural yoghurt.	

Nutrients	Nutrient analysis	NRV/goal
^∆^ DEER (Daily estimated energy requirement) (calories)	1403	2283
Carbohydrate (g) % TE	180 51	256–370 45–65
Total Sugar ^‡^ (g)	86	-
Free sugar (g) % TE	7.7 2.2	- <5%
Starch (g)	93.6	-
Protein (g) % TE	78.6 22	15–25
Fat (g) % TE	33.6 22	20–35
Saturated fat (g) % TE	11.3 7.2	≤10
Trans fats (g) % TE	0.5 0.3	2.5 <1 *
MUFA (g) % total fat	14 47	- -
PUFA (g) % total fat	4.2 14	- -
Linoleic acid (O6 PUFA) (g)	3.3	8 ^†^
Alpha-linoleic acid (O3 PUFA) (g)	0.4	0.8 ^†^
Omega-6:omega-3 ratio	8.3	10
Fibre (g)	23.2	20 ^†^
Thiamine (mg)	1.2	0.9
Riboflavin (mg)	2.1	0.9
Niacin (mg)	15.6	12
Vitamin C (mg)	31	40
Vitamin A (µg)	1696.5	600
Vitamin E (mg)	8.9	8 ^†^
Vitamin B_12_ (µg)	4.3	1.8
Folate, total (µg)	507.5	300
Calcium (mg)	1296	1000
Iron (mg)	8.06	8
Magnesium (mg)	321.8	240
Zinc (mg)	11.2	6
Sodium (mg)	1636	400–800 ^†^
Potassium (mg)	3322	2500 ^†^
Phosphorus (mg)	1601	1250
Selenium (µg)	48.3	50
Iodine (µg)	221.3	120

^∆^ DEER, Desirable Estimated Energy Requirement. The dietary energy intake predicted to maintain the energy balance and growth in healthy individuals or groups of individuals of a defined gender, age, weight, height, and level of physical activity consistent with good health and/or development. * WHO recommendation for trans fats. ^†^ AIs used as RDIs were unavailable. ^‡^ Total sugar is defined as ‘free sugar’ (monosaccharides and disaccharides added to foods and beverages by the manufacturer, cook, or consumer and sugars naturally present in honey, syrups, fruit juices, and fruit juice concentrate) [22] and ‘intrinsic sugar’ (natural sugar found in intact fruit, vegetables, and milk). AI, adequate intake; LCHF, low-carbohydrate, high-fat diet; MUFA, monounsaturated fat; NRV, nutrient reference value; PUFA, polyunsaturated fat; RDI, recommended daily intake; TE, total energy.

## 4. Discussion

Childhood and adolescence are characterised by periods of rapid physical growth. Optimal growth is facilitated by sufficient intakes of the following key nutrition components: protein and essential fatty acids and micronutrients, noting that there are limited data on the essentiality of carbohydrates and, therefore, no recommended minimum carbohydrate intake in children [18].

This work demonstrated that well-formulated LCHF meal plans designed for children can be micronutrient-, protein- and energy-replete. Some discussion containing protein, saturated fat, thiamine, and fibre is warranted.

### 4.1. Protein

In children, it is crucial to meet their protein needs to facilitate growth. In the context of our LCHF meal plans, this fundamental condition for growth can be assured.

In developing our plans, the protein intake was based on the 15–25% AMDR recommended by the Australian NMHRC and New Zealand MOH. Due to the nature of the foods that are eaten as part of a LCHF approach, each of the four meal plans slightly exceeded the 25% protein upper threshold by up to three percentage points (i.e., 28%). This is an amount that we do not consider excessive [18], noting that the upper limit for protein in children is higher than 25% in a number of comparable countries—for example, Canada, which has an upper limit of 35%.

Protein recommendations for children and adolescents are represented in two different units that are not synonymous: AMDR (percentage of total energy) and grams per kilogram of bodyweight (g/kg). We used AMDR to inform our meal plans. If converted to g/kg of bodyweight, the AMDR for protein appears to provide an excessive amount of protein (2.5–4.4 g/kg) compared to the NRV guidelines, which are 0.77–0.87 g/kg for females and 0.94–0.99 g/kg for males [22].

We note, however, that although there are no clearly defined safe upper limits for dietary protein in older children and adolescents, survey data indicate that healthy 1–3-year-old children can tolerate a daily protein intake of 5 g/kilogram bodyweight [23]. As such, we regard this as a safe upper limit for children of all ages, and our meal plans fall well within this. We also did not used any supplemental protein sources or excessive portions of animal-based protein-rich foods in our meal plans, which might ordinarily cause concern for protein excess at first glance. Despite this, we believe that further studies are needed to establish both the optimal range and the upper limit of protein for children and adolescents, particularly relating to the context of low-carbohydrate diets.

### 4.2. Saturated Fat

Our dietary plans did not meet the saturated threshold of <10% of the total energy, as recommended in the dietary guidelines; our plans exceeded the threshold by up to 2.5 times. If saturated fat is considered a concern, it is possible to formulate a nutrient- and energy-replete LCHF diet that is also low in saturated fat [16]. In our previous work [16], we created two sets of LCHF meal plans for adults (one with ad libitum saturated fat and the other with a saturated fat threshold <10% of the total energy). Both sets of meal plans were shown to be energy- and nutrient-replete. We note, however, that after all relevant coconut-based and animal foods were removed or altered to low-fat or non-fat versions (i.e., dairy products and meats), saturated fat still slightly exceeded the 10% threshold, and it was only with the reduction of foods such as avocados, seeds, olive oil, and macadamia nuts that saturated fat met the threshold guidelines. Given their powerful health benefits, whether it is prudent to reduce these foods is an important consideration. In any case, in light of the ongoing debate about the role of saturated fats in cardiovascular health and total mortality [24,25,26], it is possible that saturated fat may be less of a nutrient concern than previously thought.

It is important to note that much of the saturated fat in the meal plans we constructed came from full-fat dairy. We believe full-fat dairy products are important to include in children’s diets, as they are an excellent source of nutrients that play a vital role in childhood growth and development [27]. They may also lower the risk of diabetes [28,29] and reduce the risk of dental caries in children [30]. They may also have a protective role against childhood obesity and adiposity; however, this is still being substantiated in the literature [31,32,33,34]. A recent systematic review of 28 cross-sectional and prospective studies and one RCT in children 2–18 years old reported that full-fat dairy products are not associated with increased levels of weight gain or obesity or increased cardiometabolic risk, and the study authors stated that the dietary guidelines, which recommend that children consume reduced-fat dairy products, are not consistent with the evidence [33]. A further 2022 systematic review and meta-analysis corroborated these findings amongst cross-sectional studies but reported inconclusive evidence of this inverse relationship in prospective studies relating to being overweight, not obese. [34]. High-quality RCTs in children that compare the effects of full-fat vs. reduced-fat dairy intake on measures of adiposity and biomarkers of cardiometabolic disease risk are needed to provide more clarity in this area.

### 4.3. Thiamine and Fibre

As they are low in cereal- and grain-based foods, it is important that LCHF diets provide adequate dietary fibre and micronutrients found in these foods, particularly vitamin B1 (thiamine) [35]. We show here that the sample LCHF meal plans for children meet both fibre and thiamine requirements, a result that aligns with both our previous work on LCHF adult meal plans [16] and a recent study of micronutrient statuses in LCHF diets for overweight adolescents [14]. As shown in our meal plans, the thiamine requirements can be met through the inclusion of animal proteins, nuts, seeds, several green vegetables, and fibre through the inclusion of a variety of fruits, vegetables, nuts, and seeds.

### 4.4. A Family-Friendly Approach

When compiling diets for children and adolescents, it is vital to take into account their contexts and preferences. We designed our sample meal plans with consideration towards school rules (for example, around nuts), appropriate meal and snack timings, children’s palates and preferences, and a diet that is acceptable to, and achievable for, families. We emphasised whole foods in our approach; however, we included some commercially available products to illustrate that a predominantly whole-food diet can be supplemented with quality low-carbohydrate products that are convenient and family-friendly. Homemade choices can always be used as alternatives, if desired.

### 4.5. Comparison with the Australian Dietary Guidelines Sample Meal Plan

We reviewed the ADG sample meal plans alongside the LCHF meal plans we created. The ADG sample meal plan meets the majority of the micronutrient requirements (with the exception of selenium and vitamin C); however, it fell short on essential fatty acids and energy. The meal plan, which is intended for a lightly active 10-year-old female of healthy body size, supplied approximately 1400 kcal—61% of the child’s daily estimated energy requirement.

The ADG meal plan provides 180 g of carbohydrates, with the total sugars forming almost half of the carbohydrate intake. The ADG plan aligns with the WHO recommendation of ‘free sugars’ of being less than 5% of the total energy for optimal health, as do our plans. However, it is important to note that sugars (free or intrinsic) and starch are both carbohydrates and that, irrespective of length or complexity, sugars and starches are metabolised in the body to form one of three monosaccharides: glucose, fructose, or galactose. Therefore, consuming 180 g carbohydrates will result in the metabolic impacts of the total carbohydrate amount, in addition to the quality of the carbohydrates consumed, which evidence shows may detrimentally impact health in some populations, depending on the degree of insulin resistance, glucose intolerance, or other biological predisposition [36,37].

### 4.6. Limitations

There were several limitations to this work. First, it is a series of hypothetical meal plans and does not reflect actual intakes. Second, the average height and BMI midpoints were utilised from the World Health Organization growth charts, which were published in 2007 and may no longer be indicative of the average height and BMI of children in 2021. Finally, our micronutrient analyses were limited to the available values in the FoodWorks database and may not reflect the precise nutrient intake of the variety of products on the market.

## 5. Conclusions

Our sample LCHF meal plans demonstrated that a well-planned LCHF dietary approach can provide adequate energy, protein, and micronutrients for children and adolescents; suit children’s food preferences; and be convenient for families. Indeed, our sample LCHF meal plans provided more nutrients and energy than the sample meal plan for children endorsed by the ADG. We note that daily variations will occur as a result of the preferences of children and that further dietary restrictions as a result of culture, religion, allergies, and choice may further impact the nutritional profile, and reiterate that planning is essential to formulate a LCHF that will support healthy growth and development in children.

## Figures and Tables

**Table 1 nutrients-14-01598-t001:** Demographic data used for the LCHF case studies and for the ADG case.

	Age Range (Years)	Reference Height (cm)	Reference Weight (kg)	BMI	^†^ PAL	Energy (kcal)
Male	11 y	142 cm	34.3 kg	17	1.7	2175
Female	11 y	145 cm	35.7 kg	17	1.7	2016
Male	16 y	172 cm	60.7 kg	20.5	1.7	2968
Female	16 y	162 cm	54.5 kg	21	1.7	2442
* Female	10 y	138 cm	32.9 kg	16.7	1.6	2283

* ADG, case study; ^†^ PAL, physical activity level.

## Data Availability

Data (nutrient analysis) are unable to be placed on a data sharing system due to it being embedded in specific nutrient analysis software that is unable to be shared outside of the software programme. However, a print screen version of the data is available upon request from the author.

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
