# Peer review of "Assessing the Nutrient Status of Low Carbohydrate, High-Fat (LCHF) Meal Plans in Children: A Hypothetical Case Study Design"

_nutrients, 2022, doi:10.3390/nu14081598_

Round 1
Reviewer 1 Report
Selected comments:
It is well known that a proper diet is one of the main factors responsible for human health. The diet must be balanced correctly in terms of nutrients. It should be based on grain products, vegetables and fruits, which are primarily a source of carbohydrates. Therefore, it is essential to indicate which carbohydrates to limit/eliminate from the diet (see lines 34-36).
The introduction is not consistent with the purpose of the work. The introduction must explain why the authors deal with a given research problem.
For the study, the authors give lines 66-68: "This work assesses the macro-and micronutrient analyzes of moderately low carbohydrate meal plans for children and adolescents of healthy weights, planned with children's taste preferences and palates in mind." Children's diets should contain the right amount of all necessary nutrients. It is not a question of children's preferences, especially as children's tastes differ and change with age.
Lines 36-38. I'm afraid I have to disagree with the statement given by the authors. Please note that the references in line 36 are out of date.
In this work, the authors did not address the upper tolerable intake levels established for certain nutrients. It is also essential to pay attention to the fact that although the upper levels of consumption have not been defined so far, an excess of certain nutrients may have a negative impact on human health. For example, so far the standards for UL have not been defined for protein but attempts are made to determine its upper limit of consumption. However, it is known that high protein consumption may be associated with hypercalciuria resulting in osteoporosis, acidosis, and an increased risk of kidney stones. The excess dietary protein content in childhood contributes to an increased risk of obesity.
Author Response
Please find our responses in the word document attached.

Reviewer 2 Report
In the manuscript “Assessing the nutrient status of low carbohydrate, high-fat (LCHF) meal plans in children: a hypothetical case study design”, the Authors tried to assess the nutrient and energy status of low-carbohydrate and high fat sample meal plans for children. This is the descriptive study, which the Authors designed four meal plans for children, and adolescents with weight-stable. The Authors tried to fill the gap regarding potential risks and benefits of LCHF diets for children. The study is an important contribution to science due to different opinions related to this approach.
Generally, the manuscript provides valuable information. However, I have some remarks.
Introduction;
The Authors wrote that “the application of LCHF diets in children without weight concerns is relevant; for example, in children with type 1 diabetes whose glycemic control may benefit from reducing dietary carbohydrates”. I agree with it. However, in your opinion, using of this type of diet among healthy children with normal weight is appropriate? Especially, it is a lower carbohydrate diet than the diet recommended by the Australian Dietary Guidelines. Please, explain that.
Methods;
Why the Authors decided to assess one-day meal plans and no longer? In my opinion, longer meal plans would provide a more information and better describe quality of diet.
The Authors chose activity factor of 1.7 (light-moderate level). Therefore, I have a question. There is now talk of a problem among children and adolescents regarding physical activity, which is often very low. Therefore, have you not thought of assessing diets for low physical activity as well?
Line 23;
Should be “data were assessed” not “data was assessed”.
Keywords:
Do you need these keywords? (“nutrition and dietetics; pediatrics; and public health”). I think these are not important in your study, because they are too much general and do not indicate what exactly you are investigating.
Author Response

(The authors gave the same response as above.)

Reviewer 3 Report
The authors evaluate the macro- and micronutrient composition of an example of LCHF meal plan for children and adolescents with healthy weight. They conclude that their sample of LCHF meal plan provides adequate energy, protein and micronutrients for children and adolescents.
I have some comments and queries as follow:
- The authors calculate one day meal plan and not provide any information about mean nutritional composition of a variated plan of the diet to achieving optimal nutrient status. Whit only one day is not possibile to demonstrate that LCHF diet can be micro- and macro-nutrient replete.
Why do not the authors calculate at least a weekly plan to demostrate that “well-formulated meals and careful planning” can be feasible over time?
- As discussed by the authors, the protein and saturated fat intakes are very high and in contrast with dietary guidelines for healthy and diabetes and cardiovascular disease.
A recent review (Brouns F. Overweight and diabetes prevention: is a low-carbohydrate-high-fat diet recommendable? Eur J Nutr. 2018) concludes: “There is lack of data supporting long-term efficacy, safety and health benefits of LCHF diets. Any recommendation should be judged in this light”
Did the authors consider the long-term health implications of LCHF diets, given the relatively short duration of virtually all available studies?
Lines 58-59: “The application of LCHF diets in children without weight concerns is relevant; for example, in children with type 1 diabetes whose glycaemic control may benefit from reducing dietary carbohydrates”
The authors don’t consider the relationship between high protein intake (certain in the LCHF diets) and possible nephropathy.
Lines 174-177: “We note, however, that although there are no clearly defined safe upper limits for dietary protein in older children and adolescents, survey data indicate that healthy 1-3 year-old children can tolerate a daily protein intake of 5g/kilogram of bodyweight. As such, we regard this as a safe upper limit for children of all ages, and our meal plans fall well within this”.
Please add reference.
If we look at 1-3 y olds, literature supports the unfavourable health effect of a high protein intake in early childhood. A recent review (Arnesen EK et al. Protein intake in children and growth and risk of overweight or obesity: A systematic review and meta-analysis. Food Nutr Res. 2022) concludes that: “Based on consistent findings across cohort studies, it is probable that higher protein intake, in particular of animal origin, in children ≤18 months of age is linked to subsequent higher BMI”. Another review (Hörnell et al. Protein intake from 0 to 18 years of age and its relation to health: a systematic literature review for the 5th Nordic Nutrition Recommendations. Food Nutr Res. 2013) concludes that “A high intake of protein in infancy and young childhood thus seems to be less than optimal, and associated with increased risk of obesity later in life”.
The authors conclusion about protein intake are not supported by the literature. It is clear that future studies are needed to examine the optimal range of protein intake for children and adolescents. How do the authors define their LCHF meal plan protein adequate? Furthermore, are there any studies in children with such a high protein intake?
As discuss by the authors, saturated fat exceed the upper limit intake of 10%ET but they say it is possible to plan a meal within limits of saturated fat intake. If possible, why didn't the authors do it?
Lines 198-203: “They are also shown to be protective against 198 childhood obesity, and adiposity [27,28] may lower the risk of diabetes [29,30] and reduce 199 the risk of dental caries in children [31]. Indeed, a recent systematic review of 29 studies 200 in children 2-18 years reported that full-fat dairy products are not associated with in- 201 creased levels of weight gain or obesity or increased cardiometabolic risk, and the study 202 authors state that the dietary guidelines, which recommend that children consume re- 203 duced-fat dairy products, is not consistent with the evidence [32]”
Evidence on consumption of dairy foods and human health is contradictory. Cross-sectional studies suggested an inverse association between total dairy consumption and obesity, as the authors discuss, but prospective studies in children and adolescents are limited and no conclusive. (Babio et al. Total dairy consumption in relation to overweight and obesity in children and adolescents: A systematic review and meta-analysis. Obes Rev. 2022 ).
Lines 257-259: “As such, we believe that a LCHF dietary approach could also be considered alongside the current national guidelines as an alternative nutritional approach for optimal health for children and adolescents, and the ADG meal plans are due for review”
This sentence is not supported by this work even by the literature.
Author Response

(The authors gave the same response as above.)

Round 2
Reviewer 1 Report
Thanks to the authors for the answer.
According to the reviewer, the manuscript has not been sufficiently improved. In the reviewer's opinion, the article does not meet the requirements for scientific paper.
Reviewer 3 Report
Reviewer 3
The authors evaluate the macro- and micronutrient composition of an example of LCHF meal plan for children and adolescents with healthy weight. They conclude that their sample of LCHF meal plan provides adequate energy, protein and micronutrients for children and adolescents.
I have some comments and queries as follow:
- The authors calculate one day meal plan and not provide any information about mean nutritional composition of a variated plan of the diet to achieving optimal nutrient status. Whit only one day is not possibile to demonstrate that LCHF diet can be micro- and macro-nutrient replete.
Reviewer comment: Why do not the authors calculate at least a weekly plan to demostrate that “well-formulated meals and careful planning” can be feasible over time?
This is a good point, and is the same comment raised by Reviewer 2 above. Our response, also pasted above is as follows:
We agree with this comment that a lengthier plan would have been optimal. Our goal was to keep this paper as translational / practical as possible. In reality, children’s meals don’t change markedly from day to day to any great degree, apart from perhaps dinner. We therefore decided that it would be more valuable to vary the foods / meal ideas across each of the four different case studies to illustrate the variety and how this would fare against nutrient reference values, rather than continue to repeat the same meals each day and introduce much repetition.
In addition to this, we think that to assess whether these diets can be feasible over time – from a nutrient repletion perspective – actual dietary data needs to be captured rather than hypothetical meal plans.
Reply: Since the authors have proposed an "unusual" diet, it is necessary that they formulate at least a weekly plan that helps to have a well-formulated and varied diet. If this is not practical in the paper, it would possible to add the weekly meal plan in supplemental material and to consider in the paper the nutritional composition for weekly meal plan and not for a day.
- Reviewer comment: As discussed by the authors, the protein and saturated fat intakes are very high and in contrast with dietary guidelines for healthy and diabetes and cardiovascular disease. A recent review (Brouns F. Overweight and diabetes prevention: is a low-carbohydrate-high-fat diet recommendable? Eur J Nutr. 2018) concludes: “There is lack of data supporting long-term efficacy, safety and health benefits of LCHF diets. Any recommendation should be judged in this light”
Did the authors consider the long-term health implications of LCHF diets, given the relatively short duration of virtually all available studies?
Adult studies using LCHF have been undertaken for timeframes considered ‘long-term’ - two years in both Shai, et al. (2008). Weight loss with a low-carbohydrate, Mediterranean, or low-fat diet. https://www.nejm.org/doi/full/10.1056/nejmoa0708681 and Athinarayanan, et al. (2019) Long-Term Effects of a Novel Continuous Remote Care Intervention Including Nutritional Ketosis for the Management of Type 2 Diabetes: A 2-year Non-randomized Clinical Trial. https://www.frontiersin.org/articles/10.3389/fendo.2019.00348/full (plus many other studies associated with the Virta health trials - https://www.virtahealth.com/research#Papers ). In these trials, LCHF diets have demonstrated long term safety and positive health outcomes – acknowledging the nuanced interpretations of lipidology as described above.
Studies undertaken in children vary in length. Demol et al (reference 4: doi:10.1111/j.1651-2227.2008.01051.x.) followed their adolescents up at 12 months, following a 12-week intervention. Over this timeframe, LCHF diets were documented to be safe with no modification on renal or hepatic function, vitamin deficiencies or toxicities, or electrolyte disturbances. Krebs and colleagues (2010) reported no metabolic and cardiovascular safety risks in adolescents consuming a high protein diet for 36 weeks, in the context of a low carbohydrate approach. https://www.jpeds.com/article/S0022-3476(10)00317-3/fulltext. We have no reason to believe that in the longer term, safety issues for children would be any different to that seen in the adult studies and to the existing children studies, however these studies would be worthwhile conducting.
Reply: Low- carb diet in Shai 2008 provided about 40% TE of carbohydrates (20 g for the 2-months and immediately with a gradual increase to a maximum of 120 g per day), much more than the diet proposed by the authors.
The study of Athinarayanan, et al. (2019) was conducted on overweight/obese adults with DM2, not comparable with authors’ population (normal weight children and adolescent without diseases.
The diet in Demol et al (2009) is reduced in calories and it is for obese adolescents.
Krebs (2010) evaluated the efficacy and safety of restricted caloric diets in severely obese adolescents.
All cited references are not comparable with the authors’ population or proposed diet.
A normal caloric low carb diet obviously provides more protein and fat than restricted caloric diet and this amount has not yet been studied for safety in HEALTHY children and adolescents.
Reviewer comment: Lines 58-59: “The application of LCHF diets in children without weight concerns is relevant; for example, in children with type 1 diabetes whose glycaemic control may benefit from reducing dietary carbohydrates”. The authors don’t consider the relationship between high protein intake (certain in the LCHF diets) and possible nephropathy.
This is a similar comment as that raised by Reviewer 1 above. Our response, also pasted above is as follows:
Protein and nephropathy
Regarding kidney health in general, on balance the evidence indicates that in the context of healthy kidneys, the level of protein in the diet has no impact on kidney function. This is true for both higher carbohydrate diets and low-carbohydrate diets. Devries et al (2018). Changes in Kidney Function Do Not Differ between Healthy Adults Consuming Higher- Compared with Lower- or Normal-Protein Diets: A Systematic Review and Meta-Analysis https://pubmed.ncbi.nlm.nih.gov/30383278/ High protein diets appear to be potentially problematic in the context of existing kidney disease.
Protein and kidney stones:
A recent study indicated the risks of kidney stones in paediatric populations can come from a range of dietary and environmental factors and include high ambient temperatures (reduced body fluid via heat-induced sweating), high levels of sodium, low intakes of potassium, overweight and obesity, supplemental calcium and vitamin C, high levels of sucrose and fructose, and high protein diets. It has been documented that very high animal protein diets have been associated with an increased risk of uric acid stones. Additionally, hypercalciuria is found in 40% of kidney stone formers, idiopathic hypercalciuria is mostly classified as an inherited metabolic abnormality, with up to 75% of children with kidney stones having a family history. https://link.springer.com/article/10.1007/s00467-018-4179-9
Due to the metabolic benefits of reduced carbohydrate diets, and the fact that research to date has not found that kidney stones occur more often among those who follow low-carb or ketogenic diets for other conditions, we don’t see the mildly elevated protein (in one meal plan) being a notably high risk for kidney stones. Additionally, potassium reduces risk of kidney stones, and our meal plans are high in potassium.
Reviewer comment: Lines 174-177: “We note, however, that although there are no clearly defined safe upper limits for dietary protein in older children and adolescents, survey data indicate that healthy 1-3 year-old children can tolerate a daily protein intake of 5g/kilogram of bodyweight. As such, we regard this as a safe upper limit for children of all ages, and our meal plans fall well within this”. Please add reference.
The following reference has been added. 23. Wu, G. Dietary protein intake and human health. Food Funct 2016, 7(3): 1251-1265, doi: 10.1039/C5FO01530H
Reviewer comment: If we look at 1-3 y olds, literature supports the unfavourable health effect of a high protein intake in early childhood. A recent review (Arnesen EK et al. Protein intake in children and growth and risk of overweight or obesity: A systematic review and meta-analysis. Food Nutr Res. 2022) concludes that: “Based on consistent findings across cohort studies, it is probable that higher protein intake, in particular of animal origin, in children ≤18 months of age is linked to subsequent higher BMI”. Another review (Hörnell et al. Protein intake from 0 to 18 years of age and its relation to health: a systematic literature review for the 5th Nordic Nutrition Recommendations. Food Nutr Res. 2013) concludes that “A high intake of protein in infancy and young childhood thus seems to be less than optimal, and associated with increased risk of obesity later in life”.
The authors conclusion about protein intake are not supported by the literature. It is clear that future studies are needed to examine the optimal range of protein intake for children and adolescents.
This is the same comment raised by Reviewer 1 above. Our response, also pasted above is as follows:
The concern around high protein intakes in children and the consequent increased risk of obesity in childhood stems largely from two systematic reviews, one of which was conducted on children <18 months of age. https://www.ncbi.nlm.nih.gov/pmc/articles/PMC8861858/ In this study, 5 / 21 studies were RCTs, and authors reported null effects of high-protein intake in infants on weight gain in all six RCTs and reported methodological concerns regarding the risk of bias and small sample sizes. The probable relationship identified in the other studies is still related to children <18 months and being observational outcomes in nature, authors cannot attribute causation in this instance.
In the second systematic review https://pubmed.ncbi.nlm.nih.gov/23717219/ the age range and the type of study included - clinical trials and observational - varied. Authors concluded that the age period most sensitive to high protein intake is the first two years of life. Despite a probable association reported between protein take in young children and obesity in adolescence, authors acknowledge the methodological limitations of these studies, and that there is no upper limit firmly established.
Based on the above points, and the fact the many of these studies included use of infant formula where protein intakes markedly exceeded needs, we don’t see a parallel with the context outlined above and that of our target population or setting. We therefore see no reason to bring attention to this relationship in the manuscript, but have decided to provide additional text in line with the reviewer’s concern around no established optimal ranges and upper limits for protein in this population group (lines 194-198).
“We have also not used any supplemental protein sources or excessive portions of animal-based protein-rich foods in our meal plans, which might ordinarily cause concern for protein excess at first glance. Despite this, we believe that further studies are needed to establish both the optimal range and the upper limit of protein for children and adolescence, particularly relating to the context of low carbohydrate diets”.
Reply: The review by Arnesen includes infants and children between 4 months and 5 years of age, an age range not comparable with that of the authors’ diet. Furthermore,
The second review, as the authors reported, concludes that there is no upper limit firmly established.
Since the evidence is limited and inconclusive, the authors cannot conclude that their sample of LCHF meal plan provides adequate protein for children and adolescents.
How do the authors define “excessive portions of animal-based protein-rich foods”?
The authors did not provide the rate of animal to vegetal protein. Moreover, a one-day diet does not allow for a general assessment of the nutritional adequacy of the LCHF plan
Reviewer comment: How do the authors define their LCHF meal plan protein adequate? Furthermore, are there any studies in children with such a high protein intake?
We have acknowledged that our protein intake is at the high end of Australia and New Zealand’s upper limit – indeed, slightly above - but we do not consider it to be excessive, considering the context we have explained in the paper. Additionally, we have not used any supplemental protein sources or excessive portions of protein-rich foods. Regarding other studies, Krebs and colleagues reported no metabolic and cardiovascular safety risks in adolescents consuming a high protein diet for 36 weeks, in the context of a low carbohydrate approach. https://www.jpeds.com/article/S0022-3476(10)00317-3/fulltext
Reply: Krebs (2010) evaluated the efficacy and safety of restricted caloric diets in severely obese adolescents.
As I commented above, a normal caloric low carb diet obviously provides more protein and fat than restricted caloric diet and this amount has not yet been studied for safety in HEALTHY children and adolescents.
Reviewer comment: As discuss by the authors, saturated fat exceeds the upper limit intake of 10%ET but they say it is possible to plan a meal within limits of saturated fat intake. If possible, why didn't the authors do it?
We decided not to develop plans with restricted saturated fat for this work for two main reasons i. we felt we had already established a proof of concept in our previous paper in adults https://bmjopen.bmj.com/content/8/2/e018846, with much discussion on the issues faced when undertaking this exercise ii. as a result of the emerging evidence around the benefits of full-fat dairy for children which we discussed in some detail, and have built upon, in the manuscript.
Reply: Since the authors considered appropriate to formulate a specific plan for children, different from adults, the example of a well-formulated and adequate meal plan should be the best that can be achieved even with less saturated ones, if feasible.
Lines 198-203: “They are also shown to be protective against 198 childhood obesity, and adiposity [27,28] may lower the risk of diabetes [29,30] and reduce 199 the risk of dental caries in children [31]. Indeed, a recent systematic review of 29 studies 200 in children 2-18 years reported that full-fat dairy products are not associated with in- 201 creased levels of weight gain or obesity or increased cardiometabolic risk, and the study 202 authors state that the dietary guidelines, which recommend that children consume re- 203 duced-fat dairy products, is not consistent with the evidence [32]”
Reviewer comment: Evidence on consumption of dairy foods and human health is contradictory. Cross-sectional studies suggested an inverse association between total dairy consumption and obesity, as the authors discuss, but prospective studies in children and adolescents are limited and no conclusive. (Babio et al. Total dairy consumption in relation to overweight and obesity in children and adolescents: A systematic review and meta-analysis. Obes Rev. 2022 ).
Thank you for drawing our attention to the study, which alongside substantiating our text, does present a more contradictory picture. As such we have included this as a reference and have amended this paragraph to reflect this lack of consensus relating to dairy and overweight and obesity.
“It is important to note that much of the saturated fat in the meal plans we constructed comes from full-fat dairy. We believe full-fat dairy products are important to include in children’s diets as they are an excellent source of nutrients that play a vital role in childhood growth and development [27]. They may also lower the risk of diabetes [28, 29] and reduce the risk of dental caries in children [30]. They may also have a protective role against childhood obesity, and adiposity, however this is still being substantiated in the literature [31, 32, 33, 34]. A recent systematic review of 28 cross-sectional and prospective studies and one RCT in children 2-18 years reported that full-fat dairy products are not associated with increased levels of weight gain or obesity or increased cardiometabolic risk, and the study authors state that the dietary guidelines, which recommend that children consume reduced-fat dairy products, is not consistent with the evidence [33]. A further 2022 systematic review and meta-analysis corroborated these findings amongst cross-sectional studies but reported inconclusive evidence of this inverse relationship in prospective studies relating to overweight, not obesity. [34]. High-quality RCTs in children that compare the effects of full-fat vs reduced-fat dairy intake on measures of adiposity and biomarkers of cardiometabolic disease risk are needed to provide more clarity in this area”.
Reviewer comment: Lines 257-259: “As such, we believe that a LCHF dietary approach could also be considered alongside the current national guidelines as an alternative nutritional approach for optimal health for children and adolescents, and the ADG meal plans are due for review”
This sentence is not supported by this work even by the literature
We have removed this sentence and reworked the paragraph as follows to better align with our work (line 277-285).
“Our sample LCHF meal plans demonstrate that a well-planned LCHF dietary approach can provide adequate energy, protein and micronutrients for children and adolescents, suit children’s food preferences, and be convenient for families. Indeed, our sample LCHF meal plans provide more nutrients and energy than the sample meal plan for children endorsed by the ADG. We note that daily variation will occur as a result of the preferences of children and that further dietary restrictions as a result of culture, religion, allergies, and choice may further impact the nutritional profile, and reiterate that planning is essential to formulate a LCHF that will support healthy growth and development in children”.